# Molecular Mechanisms of N-Acetylcysteine in RSV Infections and Air Pollution-Induced Alterations: A Scoping Review

**DOI:** 10.3390/ijms25116051

**Published:** 2024-05-31

**Authors:** August Wrotek, Artur Badyda, Teresa Jackowska

**Affiliations:** 1Department of Pediatrics, The Centre of Postgraduate Medical Education, 01-813 Warsaw, Poland; tjackowska@cmkp.edu.pl; 2Faculty of Building Services, Hydro- and Environmental Engineering, Warsaw University of Technology, 00-653 Warsaw, Poland

**Keywords:** acetylcysteine, respiratory syncytial virus, air pollution, environmental pollution, particulate matter, respiratory hypersensitivity, molecular pathology

## Abstract

N-acetylcysteine (NAC) is a mucolytic agent with antioxidant and anti-inflammatory properties. The respiratory syncytial virus (RSV) is one of the most important etiological factors of lower respiratory tract infections, and exposure to air pollution appears to be additionally associated with higher RSV incidence and disease severity. We aimed to systematically review the existing literature to determine which molecular mechanisms mediate the effects of NAC in an RSV infection and air pollution, and to identify the knowledge gaps in this field. A search for original studies was carried out in three databases and a calibrated extraction grid was used to extract data on the NAC treatment (dose, timing), the air pollutant type, and the most significant mechanisms. We identified only 28 studies conducted in human cellular models (*n* = 18), animal models (*n* = 7), and mixed models (*n* = 3). NAC treatment improves the barrier function of the epithelium damaged by RSV and air pollution, and reduces the epithelial permeability, protecting against viral entry. NAC may also block RSV-activated phosphorylation of the epidermal growth factor receptor (EGFR), which promotes endocytosis and facilitates cell entry. EGFR also enhances the release of a mucin gene, MUC5AC, which increases mucus viscosity and causes goblet cell metaplasia; the effects are abrogated by NAC. NAC blocks virus release from the infected cells, attenuates the cigarette smoke-induced shift from necrosis to apoptosis, and reverses the block in IFN-γ-induced antiviral gene expression caused by the inhibited Stat1 phosphorylation. Increased synthesis of pro-inflammatory cytokines and chemokines is induced by both RSV and air pollutants and is mediated by the nuclear factor kappa-B (NF-κB) and mitogen-activated protein kinase (MAPK) signaling pathways that are activated in response to oxidative stress. MCP-1 (monocyte chemoattractant protein-1) and RANTES (regulated upon activation, expressed and secreted by normal T cells) partially mediate airway hyperresponsiveness (AHR), and therapeutic (but not preventive) NAC administration reduces the inflammatory response and has been shown to reduce ozone-induced AHR. Oxidative stress-induced DNA damage and cellular senescence, observed during RSV infection and exposure to air pollution, can be partially reversed by NAC administration, while data on the emphysema formation are disputed. The review identified potential common molecular mechanisms of interest that are affected by NAC and may alleviate both the RSV infection and the effects of air pollution. Data are limited and gaps in knowledge include the optimal timing or dosage of NAC administration, therefore future studies should clarify these uncertainties and verify its practical use.

## 1. Introduction

N-acetylcysteine (NAC) is a mucolytic drug, also used in the treatment of acetaminophen overdose, which has been listed by the World Health Organization as an essential medicine [1,2]. The route of administration of NAC depends on the condition that is being treated; the modes of administration of NAC include oral administration, inhalation, and intravenous injection; the therapeutic use already registered is broad, with a number of ongoing trials for various conditions, mainly focusing on pulmonary, cardiovascular, neurodegenerative, liver, and infectious diseases, psychiatric conditions, or gastrointestinal illnesses [2]. 

The molecular mechanisms of action are related to the structure of NAC: it is a thiol, a precursor of L-cysteine and contains a functional sulfhydryl group (–SH), an acetyl group (–COCH_3_), and an amino group (NH_2_) [2]. NAC has mucolytic properties due to its ability to cleave disulfide bridges that link mucus glycoproteins, but NAC also exerts anti-inflammatory effects by suppressing the nuclear factor kappa-B (NF-κB) activation with a subsequent decrease in the production of proinflammatory cytokines, and antioxidant effects [2]. Direct antioxidant actions stem from the free thiol group, which reacts with reactive oxygen species (ROS) or reactive nitrogen species, namely hydroxyl radical (^•^OH), radical anion superoxide (O_2_^•−^), hydrogen peroxide (H_2_O_2_), nitrogen dioxide (^•^NO_2_), nitric oxide (NO^•^)/nitroxyl (HNO), carbon trioxide ion (CO_3_^•−^), and peroxynitrite (ONOO^−^). NAC also has indirect oxidant effects, since it is a precursor of L-cysteine, which is needed for the biosynthesis of glutathione (GSH), which in turn maintains cellular redox status by a direct antioxidant activity and indirectly as a substrate for several antioxidant enzymes. Low intracellular levels of L-cysteine inhibit GSH synthesis [2,3]. NAC restores the intracellular supply of cysteine, thereby increasing the glutathione-to-glutathione disulfide (GSH/GSSG) ratio [2,3].

Respiratory syncytial virus (RSV) is one of the major etiological factors in acute respiratory tract infections in children, characterized by a high incidence, especially in infancy and early childhood; the vast majority of children are infected with RSV at least once during the first 3 years of life, with the RSV-specific antibody seroprevalence exceeding 90% [4]. The disease severity may vary from mild infections treated in the community to hospitalized cases of lower respiratory tract infection (LRTI), including bronchitis, bronchiolitis, and/or pneumonia, and some patients may require intensive care [4,5]. Exposure to air pollution, including cigarette smoke, particulate matter, benzene, ozone (O_3_), carbon monoxide (CO), or nitric dioxide (NO_2_), appears to be associated with a higher RSV incidence and disease severity, resulting in a greater socioeconomic impact of the disease [6,7,8,9,10,11,12,13]. The World Health Organization has announced a list of the six most important air pollutants, including particulate matter ≤10 µm in diameter (PM10), particulate matter between 0.1 and 2.5 µm in diameter (PM2.5), NO_2_, O_3_, sulfur dioxide (SO_2_), and CO [14]. The data on the exact pathomechanisms involved in RSV and air pollution are rather scarce, but the main conclusions focus on decreased host antiviral defenses, facilitating viral entry and higher viral load, the increased virus release from primarily infected cells, and the prolonged and/or increased inflammatory response which may lead to histopathological changes, enhanced mucus secretion, and AHR; one of the most important common mechanisms seems to be mediated, at least in part, by oxidative stress [15]. 

Thus, a combination of the anti-inflammatory, antioxidant, and mucolytic activity is desirable in the treatment of various respiratory diseases, and the clinical use of NAC has been extensively investigated in chronic obstructive pulmonary disease, idiopathic pulmonary fibrosis, or cystic fibrosis [2,16,17,18]. In infectious diseases, NAC has been shown to inhibit the replication of human influenza virus in lung epithelial cells, human immunodeficiency virus (HIV) and RSV and to reduce the production of pro-inflammatory cytokines, including IL-8, CXCL10, CCL5, and IL-6 [19,20]. After the SARS-CoV-2 pandemic began, the NAC properties and the role of cytokine storm in the pathophysiology of COVID-19 gave an impetus to start a number of clinical trials on the treatment of SARS-CoV-2 [21,22]. 

The rationale to perform this scoping review was increasing evidence of an association between the RSV morbidity and air pollution. Therefore, we aimed to review the existing evidence on the molecular mechanisms of NAC that could be targeted in the course of RSV infection and, at the same time, would be mediators of the adverse effects of air pollution. 

## 2. Material and Methods

### 2.1. Study Protocol

The Preferred Reporting Items for Systematic reviews and Meta-Analyses extension for Scoping Reviews (PRISMA-ScR) was used to prepare the study protocol, which may be obtained from the corresponding author; due to the nature of a scoping review, it is a common practice not to register the study protocol prior to conducting the study, nor is it required. The PRISMA-ScR checklist is available in Appendix A. The preparation of the study included the conceptualization of the topic, which aimed to specify the areas of interest to be searched for, the eligibility criteria, the search strategy (adapted for different databases), the extraction methods, and a method of presenting the synthesis of the results. 

#### 2.1.1. Conceptual Framework

The aim of this study was to investigate the mechanisms of NAC action in RSV infection and air pollution. Data on this topic, especially studies combining the analysis of RSV and air pollution, are expected to be scarce, so the search strategy included identification of the mechanism and verification of its potential usefulness in relation to RSV and air pollution. As presented previously, an interaction between RSV and air pollution, which potentially enhances the pathological effects of both infection and pollution, may be divided into 3 main areas of interest, and so are the results presented. The directions of interaction are: (1) a facilitated viral entry (caused by the enhanced RSV adherence or damage to the host protective barriers), (2) an increased viral load (due to the increased RSV virulence, or the increased release from infected cells, or the decreased host immune response that impedes virus clearance), and (3) an exaggerated host response (a prolonged or inadequate inflammatory response that may lead to hyperreactivity reactions and histopathologic changes) [15]. We aimed to generalize the results in order to make it easier for the reader to understand the possible pathomechanisms involved, but it must be emphasized that: 1. the pathomechanisms present in humans may differ from those studied in animal or cell models, although the mechanisms are expected to be common and the choice of models is dictated by the potential reflection of those present in humans; 2. the mechanisms may differ significantly between the pollutants, so the pollutant studied is mentioned in the table that describes the included publications; 3. the mechanisms may vary with time, dose, or situation, e.g., in the majority of cases, the type of response would be stopped in a relatively short period of time, although in some particularly susceptible populations, it may be prolonged (e.g., patients with asthma); 4. the direction and magnitude of the mechanism activation or deactivation may vary, and the same mechanism may be activated or blocked, depending on the stage of infection or response to infection; and 5. studies demonstrating the effects of NAC have mostly been performed in cellular or animal models, and the effects in humans may vary. Detailed knowledge of the doses may also be crucial for switching on/off the specific pathomechanisms. 

#### 2.1.2. Eligibility Criteria

Only original papers published within the last 25 years were eligible. There were no restrictions in terms of the language of publication (in the case of papers published in languages other than English, Spanish, German, or Polish, translation assistance was assumed), study design, including the choice of model (i.e., human cellular, animal, or mixed) or study setting, but we only accepted peer-reviewed papers. The main reason for the above restrictions was to obtain high-quality, recent data and to eliminate possible discrepancies related to a change in method sensitivity. The main inclusion criterion was an analysis of an (at least one) NAC mechanism in RSV infection or air pollution. We excluded studies that did not analyze the mechanism of the NAC action, focused on the NAC mechanism in conditions other than those analyzed (i.e., RSV infection or air pollution or substances that are not used as air pollution models), analyzed only maternal/in utero exposure, were epidemiological studies, or assessed an interaction between air pollution and RSV without examining the NAC effects. If a study focused only on combined effects (air pollution and another pathogen, e.g., influenza or rhinovirus, or an allergy model), the study was excluded. Because of the expected wide range of possible effects of pollutants on organs, we also excluded studies that focused on pathophysiology beyond the respiratory tract; while studies using the cells present in the respiratory tract (epithelial cells, lymphocytes, macrophages, monocytes, etc.) were accepted, we excluded those that focused on cells not commonly associated with the respiratory tract or that focused on other organs. The detailed inclusion and exclusion criteria are shown in Table 1. 

#### 2.1.3. Search Strategy

A systematic search of three bibliographic databases, Scopus, PubMed (Medline), and Cochrane, was executed, searching for relevant articles published after 1998; the last search was performed on 7 September 2023. Four different categories were considered in the search strategy: 1. N-acetylcysteine, 2. molecular mechanism of action, 3. RSV, 4. air pollution. The Boolean operators AND and OR were then used to combine the search results as follows “1” AND “2” AND (“3” OR “4”). Pubmed was used for the primary search, which allowed us to generate a list of MeSH terms with their headings and subheadings, as well as the list of translations; in addition, the search criteria were converted to meet the requirements of other databases (the search strategy is shown in the appendix in the Appendix A). Sources of information were not limited to the database; although review articles were excluded from the presentation in this paper, they were screened for potentially relevant (but omitted in the database search) original papers; the list of references in each included article was also checked. We searched for peer-reviewed papers only, so conference communications (in any form), preprints, or unaccepted papers were not sought, nor was gray literature search included in the eligibility criteria.

The selection of sources of evidence: Two authors (A.W. and T.J.) independently performed a bibliographic database search; first, titles/abstracts were screened for inclusion/exclusion criteria, then the full text was critically appraised. All screening results were compared and, in case of any doubt, consensus was reached among all authors. 

#### 2.1.4. Data Extraction

A calibrated form (extraction grid) was used to extract data. The extraction was performed independently by the authors, and a final abbreviated version of the extraction grid (approved by consensus) is shown in Table 2. The extraction grid included information on the first study author, publication title, site where the study was conducted, journal name, and year of publication, as well as detailed information on the study, regarding the model used in the study (human, animal, or cellular), the NAC treatment (dose and treatment timing—pretreatment versus treatment during or after exposure), the air pollutant type, and the virus exposure; the grid also provided the most significant results on mechanisms. If important data were missing, an attempt was made to contact the authors (according to the contact details published in the article) to obtain these data. There was no simplification of the areas of interest analyzed. We did not perform a critical appraisal of the studies because of the expected high quality of the papers, guaranteed by both the generally challenging topic and the review process; in a previously published analysis of the interaction mechanisms between RSV and air pollution, we also found a very high quality of the papers, which did not raise doubts about their scientific soundness [15].

#### 2.1.5. Synthesis of the Results

The general results are presented in Appendix A, which summarizes the characteristics of the included papers (details of the study model, the type of exposure (RSV and/or type of air pollutant), the mechanism analyzed, and the main results) and the narrative synthesis, focusing on the previously mentioned tripartition of: (1) viral entry, (2) increased viral load, and (3) exaggerated host response to facilitate the understanding of the actions and mechanisms of NAC at different stages of infection [15]. Also, due to the complicity of the subject at each point, a relevant comment and a brief explanation of the possible role of the mechanism had to be made. In the Discussion Section, we also presented the most relevant aspects from a clinical point of view: a brief comment on NAC dosage and clinical trials conducted in humans. The diagrams summarize the main findings of the review, with the aim to simplify and illustrate the mechanisms involved.

## 3. Results

### 3.1. Search Results

A systematic database search yielded 167 papers, including 13 duplicates, which were excluded from further analysis. The title and abstract screening excluded 122 of the 154 publications (54 due to lack of an appropriate model, 50 studying NAC mechanism in condition other than RSV or air pollution, 16 due to lack of analysis of NAC/NAC molecular mechanisms, 2 due to only maternal/in utero exposure analysis), and 32 full-text articles were retrieved and assessed for eligibility. Sixteen studies did not meet the inclusion criteria because of a focus on a disease other than RSV or air pollution (*n* = 10), analysis of a substance other than NAC (*n* = 3), and the lack of analysis of NAC mechanisms (*n* = 3). The reference list search yielded a further 12 studies, and finally, 28 papers were included in the analysis; the PRISMA flowchart is shown in Figure 1.

### 3.2. Study Characteristics

Twenty-eight papers published between 1998 and 2023 were included in the final analysis; thirteen studies (46.4%) were published in the last decade (2013–2023), and the majority of studies (20 studies, 71.4%) were published in the last 15 years (Figure 2). We did not observe an increasing trend; the number of published studies accumulated steadily. The majority of studies were conducted in the USA (*n* = 9), followed by China (*n* = 6), multisite studies (*n* = 5, including four studies in the USA, three in China, and one in Japan, Mexico, Poland, Sweden, UK), Spain (*n* = 3), and single studies in the remaining countries (Malaysia, the UK, South Korea, Japan, and Australia). 

There were sixteen research studies focusing on air pollution and twelve on RSV infection, including three studies assessing the effects of NAC on both RSV infection and air pollution. The most commonly investigated air pollutant was TiO_2_-NP (five studies), followed by cigarette smoke (four studies), diesel exhaust (three studies), and ambient particles (three studies, including particulate matter, PM2.5 or concentrated ambient particles). In general, a strong preference for human cellular models was observed (18 studies), especially for modelling RSV infection. Three studies (one investigating the mixed effect of RSV and air pollution) were performed in both human and animal models, and seven studies included animal models (four mouse and three rat models) used to assess air pollution mechanisms; however, the majority of studies (nine studies) on air pollution effects were performed using human cellular models (Figure 3).

### 3.3. Molecular Mechanisms

#### 3.3.1. Facilitated Viral Entry

Viral entry could be facilitated by two main mechanisms: (1) impaired protective barriers (either due to a damage to the existing barrier or the presence of additional transport channels), or (2) an increased viral adsorption (due to changes in the cell surface that promote viral adherence, or the RSV gaining exceptional virulence that would facilitate its adherence to the cell surface or entry) [15]. No information was found on increased RSV virulence per se, but viral entry may be facilitated by the mechanisms shown below.

Epithelial barrier

The function of epithelium may be compromised by nanoparticles; Smallcombe et al. demonstrated in cellular (human bronchial epithelial cells, HBECs) and murine models that titanium dioxide nanoparticles (TiO_2_-NP) enhance RSV-induced disintegration of the epithelial barrier by disrupting intercellular complexes (apical junctional complexes, AJC) and remodeling the intracellular anchoring of apical junctional complex (AJC) with the cytoskeletal actin [39]. As a result, the epithelium becomes “leaky” instead of being an impermeable barrier, and viral infection is enhanced [39]. A treatment with NAC diminished the disassembly of AJC, improved the epithelial permeability and reduced the extent of RSV infection as assessed by lowered viral titers [39]. The authors highlight the role of ROS production following co-exposure to RSV and TiO_2_-NP, as intracellular ROS levels increased rapidly and remained high for up to 36 h [39]. ROS may also mediate an increase in ICAM-1 (intercellular adhesion molecule- 1, CD54) expression [35] (Figure 4).

The role of ROS was also underlined by Wen et al., who compared the influence of non-cytotoxic concentrations of different nanoparticles on the adherens junction in a cell model of human umbilical vein endothelial cells (HUVECs). Interestingly, no variations in ROS generation were detected after the exposure to TiO_2_-NP, although there was an increase in catalase (CAT) activity, which is a marker of ROS upregulation [45]. Nevertheless, detrimental effects on the endothelial integrity were observed: a number of intercellular gaps and an increased permeability of the endothelial layer were detected, together with blurring of the VE-cadherin (an adherens junction protein) on the cell surface, accompanied by a decreased VE-cadherin expression throughout the cell [45]. The use of NAC reversed both the oxidative stress and the gap formation, suggesting its critical role with regard to oxidative stress [45]. 

ICAM-1

The glycoprotein (G) and fusion protein (F) play a crucial role in RSV attachment and entry into the cell; while the G protein binds to a receptor on the cell surface, the F protein interacts with the cell membrane, and its transition from pre- to post-fusion form allows viral entry [50]. ICAM-1 has been proposed as one of the F protein receptor candidates [51]. ICAM-1 is a glycoprotein constitutively present on the surface of various cells, including epithelial and endothelial cells, lymphocytes, and monocytes; an upregulation in response to specific stimuli, such as injury or infection, is mediated by cytokines (TNF-α, IFN-γ) or ROS [52,53]. The RSV infection induces a time- and dose-dependent increase in ICAM-1 expression on human endothelial cells. The upregulation is mediated by protein kinase A and C, phosphoinositide 3-kinase (PI3K), and p38 mitogen-activated protein kinase (p38 MAPK) [54]. A significant role in the synthesis of ICAM-1 mRNA has been shown for the transcription factors- NF-κB and CCAAT-enhancer-binding proteins (C/EBP) [55]. 

As a result of RSV stimulation, polymorphonuclear neutrophil granulocytes (PMN) express higher adherence and transmigration rate, leading to an increased PMN influx into the bronchoalveolar spaces [54]. Notably, RSV-infected cells overexpress ICAM-1, and anti-ICAM-1 antibodies have been proposed to ensure the delivery of drug-containing nanoparticles directly to the infected cells [56,57,58]. 

Results regarding the influence of NAC on ICAM-1 expression are somewhat contradictory, but this contradiction may be illusory. A study by Mata et al. demonstrated that ICAM-1 expression in primary normal human bronchial epithelial cells (NHBEC) increased when they were cultured with RSV, but this increase could be abolished by NAC pretreatment [35]. However, Modestou et al. showed that cigarette smoke extract inhibited IFN-γ-induced ICAM-1 expression in human tracheobronchial epithelial (hTBE) cells, but NAC (and glutathione monoethyl ester, GSH-MEE) was able to alleviate the inhibitory effect [36]. ICAM-1 plays two distinct roles in RSV pathogenesis: it may be a receptor for RSV, but it also mediates the immune response. Hence, while ICAM-1 may be critical in the early stages of infection as part of an increased viral adsorption, it also induces an antiviral response in later stages of infection, contributing to a decrease in viral titers. 

In addition to ICAM-1, other surface molecules may also be induced by air pollution; TiO_2_-NP induce the expression of ICAM-1, but also VCAM-1, PECAM-1, and selectins (E and P) in the endothelium, as well as their ligands on monocytes, including both receptors for early (E-selectin ligand (sLe^x^) and P-selectin ligand (PSGL-1)) and late adhesion molecules (ICAM-1 ligand (LFA-1),VCAM-1 ligand (VLA-4), and PECAM-1 ligand (αVβ3)) [38,59]. NAC inhibits the expression of surface receptors [38]. The adhesion of monocytes to endothelial cells was increased to the same extent when either or both (i.e., monocytes and endothelium) were exposed to TiO_2_-NP [38]. Interestingly, blocking the ICAM-1 ligand (LFA-1) in the RSV infection model prevents the neutrophil adhesion to RSV-infected cells, but does not reduce the neutrophil influx or viral load, whereas blocking the VCAM-1 ligand (VLA-4) has the ability to prevent eosinophilia and airway hyperresponsiveness in the mouse lung tissue model [60,61]. 

EGFR

Another proposed mechanism of RSV cell entry involves phosphorylation and activation of EGFR and the EGFR-related cascade of signaling factors (PI3K, PKC, Cdc42, PAK1, and N-Wasp) [62]. RSV activates phosphorylation of EGFR and its downstream signaling cascade once bound to the cell surface, causing actin rearrangements and endocytosis by micropinocytosis [62]. However, a depletion of EGFR or blockade of EGFR signaling factors by corresponding inhibitors impairs endocytosis and reduces the RSV infection [62]. The authors found that the F protein requires activation by a second proteolytic cleavage that occurs after internalization, and a removal of an F1 subunit peptide resulted in RSV gaining infectiousness [62]. In this context, NAC may block the phosphorylation of EGFR in HBECS, thereby preventing endocytosis and possibly the infectivity of the virus [25,62]. 

#### 3.3.2. Increased Viral Load

An increase in the viral load may origin from an impaired host antiviral response, a facilitated RSV spread after the infection onset (by a facilitated release from the primarily infected cells), or due to the RSV gaining exceptional virulence. 

##### Decreased Antiviral Response

IFN-γ antiviral effects

Human T cells and natural killer (NK) cells have the ability to produce IFN-γ, which binds to its receptors and causes an activation of the JAK/STAT (Janus kinase/signal transducer and activator of transcription) pathway by phosphorylating Stat1; the dimerized phosphorylated Stat1 then translocates to the nucleus and activates the antiviral genes [36,63]. Cigarette smoke extract (CSE) inhibits the phosphorylation of Stat1 and inhibits the IFN-γ-mediated decrease in RSV mRNA and the RSV protein expression [36] (Figure 5). An experimental model in hTBE cells by Modestou et al. demonstrated that the use of N-acetylcysteine (or GSH-MEE) is able to restore the antiviral effects of IFN-γ [36]. A significant part of the effects of cigarette smoke on type II IFN signal transduction is mediated by ROS, which will be discussed below [36,64,65]. 

##### Facilitated Virus Release

A shift from apoptosis towards necrosis and ROS generation

The higher viral load observed in hTBE cells exposed to CSE and RSV appears to be related to the potential of cigarette smoke to prevent RSV-induced caspase activation [27]. In cells challenged with CSE and RSV, necrosis is more pronounced than apoptosis, which leads to a virus release from the infected cells, as opposed to apoptosis, which is not only a more efficient method of removing infected cells, but also causes less inflammation [27].

An experimental model by Groskreutz et al. involved the use of specific inhibitors of various components of cigarette smoke, i.e., reactive aldehydes (aldehyde dehydrogenase), nitric oxide (LNAME), intracellular ROS (PEG-superoxide dismutase and PEG-catalase) and extracellular ROS (superoxide dismutase and catalase), and NAC, which non-specifically inhibits various components of cigarette smoke, including reactive aldehydes, ROS, and nitric oxide [27]. 

A series of experiments showed that the use of NAC and aldehyde dehydrogenase was able to partially reverse the shift from apoptosis to necrosis of the epithelial cells, suggesting that ROS and acrolein play a significant role in this complex system [27]. This is in line with the previous studies showing that acrolein promotes necrosis rather than apoptosis by a number of different mechanisms: acrolein has the ability to deplete cellular glutathione, bind to its derivatives, alkylate caspases, and reduce mitochondrial ATP synthesis; ROS are also capable of depleting glutathione and altering ATP synthesis [66,67,68,69,70]. It has been shown in a cellular model that RSV without cigarette smoke exposure also generates ROS and activates the AMP-activated protein kinase/mammalian target of the rapamycin (AMPK-MTOR) signaling pathway, thereby inducing autophagy and promoting viral infection [31]. Knockdown of autophagy genes (e.g., *beclin1*) or treatment with an AMPK inhibitor (compound C) or NAC was able to inhibit the RSV replication, but while compound C was able to block autophagy-dependent RSV replication, NAC showed the ability to inhibit both the autophagy-dependent and autophagy-independent pathways [31]. 

Certainly, other mechanisms are also involved in the induction of autophagy at the expense of a reduced apoptosis; Chakraborty et al. showed that RSV-infected HBECs are more susceptible to RSV infection after an exposure to TiO_2_-NP, and this relationship is mediated by an upregulation of the nerve growth factor (NGF) and its TrkA receptor [71]. In contrast, silencing of the NGF gene expression or inhibition of the *beclin-1* (one of the autophagy genes mentioned above) resulted in an increased apoptosis and a decreased viral load [71]. 

From a clinical perspective, the study by Zhou et al. combined the assessment of apoptosis markers with the clinical perspective; in the group of 126 elderly patients with chronic obstructive pulmonary disease (COPD), three types of treatment were implemented: oral NAC, inhaled terbutaline sulfate, and a combination of both [72]. All treatment protocols resulted in a decrease in Fas receptor/apoptosis antigen 1 (Fas/APO-1) and soluble Fas (sFas) levels, as well as an improvement in the redox status, i.e., a decrease in ROS and a decrease in an indicator of oxidative stress, malondialdehyde (MDA), accompanied by an increase in the superoxide dismutase (SOD) and the glutathione peroxide enzyme (GSH-PX), and the best results were obtained when the two drugs were combined [72].

#### 3.3.3. Exaggerated Host Response

##### Exaggerated Proinflammatory Response

Production of proinflammatory chemokines and cytokines 

A number of proinflammatory chemokines and cytokines have been suggested to be responsible for the exuberant immune response after the co-exposure to an RSV infection and air pollution, but the most extensive data underscore the role of interleukin-6 (IL-6), IL-8, IL-17, TNF-α, monocyte chemoattractant protein-1 (MCP-1), and, regulated upon activation, normal T cell expressed and secreted (RANTES) [15]. 

The pathogenesis and clinical picture of the RSV disease course is highly dependent on the inflammatory response, and the study by Lei Chi et al. demonstrated anti-inflammatory effects of NAC: both the mRNA expression and the cytokine concentrations (TNF-α, IL-6, IL-1β, and IL-18) in the cell supernatant were significantly reduced after NAC treatment [25]. Similarly, a study by Mata et al. showed that NAC suppressed the expression of IL-6, IL-8, and TNF-α [19]. An earlier study by Dick focused on the induction of TNF-α by different types of ultrafine particles (carbon black, cobalt, nickel, and Ti-02) and showed that the addition of NAC (or GSH-MEE) blocked the release of TNF-α from rat alveolar macrophages [26] (Figure 6).

In a murine model of benzo[a]pyrene-induced acute lung injury, the ability of NAC pretreatment to attenuate the inflammatory response was demonstrated; the gene expression of *TNF-α*, *IL-6*, *MCP-1*, and *MIP-2* (macrophage inflammatory protein 2, *Cxcl2)* was decreased [49]. An early study by Mastronarde et al. (1998) suggested that NAC (and two other antioxidants, DMSO and DMPO) were able to block RSV-induced IL-8 production; the exact pathways and the role of oxidant-sensitive transcription factors (NF-κB, activator protein-1-AP-1, and NF-IL6) were discussed, as the antioxidants did not inhibit the binding of NF-κB, but did inhibit the binding of AP-1 and NF-IL6 [34]. A detailed insight into IL-8 production induced by RSV and cigarette smoke concentrate was provided by Castro et al.; the expression of IL-8 gene and protein was increased after the co-exposure and NF-κB bound to the site in the IL-8 promoter [73]. A major role has been suggested for the interferon stimulatory response element (ISRE) site of the IL-8 promoter: This site is activated by IRF-1 and IRF-7, activation of which is, in turn, enhanced after co-exposure, leading to an exuberant and prolonged inflammatory response [73]. The IL-8 mRNA stability and gene transcription are increased after NO stimulation, and while NAC did not affect IL-8 levels, other antioxidants (DMSO and DMTU) inhibited NO-induced an IL-8 transcription, suggesting that hydroxyl radicals, rather than intracellular glutathione redox status variations, may be involved [40]. Of note, an increased IL-8 expression and an IL-8 gene induction after a PM2.5 exposure appears to be mediated not only by oxidative stress, but also by endocytosis [48]. A HBECs line model (BEAS-2B cells) was used to measure the IL-8 expression, and pretreatment with NAC significantly decreased the IL-8 expression [48]. With regard to endocytosis, the addition of an endocytosis inhibitor (cytochalasin D) resulted in a 60% reduction in IL-8 levels, indicating that endocytosis is also involved in the induction of the inflammatory response, probably due to its role in the PM2.5 uptake by endothelial cells [48]. 

Contradictory results were presented by Vaughan et al. in the study of the effects of diesel emissions on primary human bronchial epithelial cells (pHBEC) obtained from COPD and non-COPD donors (but not from disease-free lungs) [41]. A variety of effects of diesel exhaust on pHBEC were observed, including an increased IL-8 gene expression and secretion, an increased cytochrome P450 1a1 (CYP1a1) mRNA expression, and a suppressed superoxide dismutase-1, but a pretreatment with NAC only reversed the latter effect, and a trend toward reduced IL-8 levels was reported [41].

The effects of NF-κB stimulation

Carpenter demonstrated a role for NF-κB and a particular kinetic pattern of proinflammatory cytokine induction: while IL-8 and MCP-1 are rapidly induced after an RSV infection, RANTES shows a rapid but delayed increase [24]. NAC suppresses RSV-induced chemokine expression in a manner different from TNF-α, suggesting that a redox-sensitive NF-κB pathway is involved in the induction of IL-8, MCP-1, and RANTES [24]. RSV induces NF-κB, which leads to an increased production of cytokines that contribute significantly to the clinical effects of an RSV infection; the induction of NF-κB in A549 airway epithelial cells is mainly mediated by the induction of the p65 (Rel A) signaling pathway, and NAC is able to inhibit the effects of NF-κB stimulation [24]. 

NF-κB may be crucial under air pollution exposure: HBECs (BEAS-2B), normal human bronchial epithelial (NHBE), and chronic obstructive pulmonary disease (COPD) human bronchial epithelial (DHBE) cells, exposed to PM2.5, exhibited a dose-dependent increase in both the binding activity and the secretion of NF-κB target cytokine [74]. Moreover, there was an increase in the cytosolic production of ROS, which activated the nuclear erythroid 2-p45-related factor 2 (NRF2) signaling pathway, whereas a prolonged and repeated exposure to PM2.5 resulted in a partial inactivation of the NRF2 pathway; a persistent mitochondrial dysfunction with a decreased cellular energy supply was observed as a consequence of a disrupted mitochondrial redox homeostasis [74]. The nuclear factor (erythroid-derived 2)-like 2 (NRF2)/Kelch-like ECH-associated protein 1 (KEAP1) is a key regulator of the antioxidant response; under oxidative stress, KEAP1 releases NRF2, instead of binding it, leading to its proteosomal degradation, and as an effect, NRF2 target genes are induced and the antioxidant response is initiated [75,76,77]. One of the target genes is the heme oxygenase 1 (HO1) gene. The study by Mata et al. in the human pulmonary epithelial cell line A549 describes antioxidant mechanisms induced by NAC action; while the HO1 expression was decreased on day 15 post RSV infection, it was not only restored, but increased after NAC pretreatment [35]. However, NRF2 itself exhibits a complicated time-dependent response; its expression is rapidly induced by day 4, with a subsequent decrease to finally reach levels lower than those in controls by day 15 post-infection. While lower doses of NAC (1 mM NAC) reverse this effect (i.e., the decrease at day 15), higher (10 mM) doses of NAC kept the NRF2 expression elevated throughout the experimental period [35]. Consequently, the total antioxidant status (TAS), which is the ability of a cell to respond to oxidative events, was decreased after an RSV infection, and was restored by a NAC pretreatment at a dose of 1 mM, and increased after a 10 mM NAC pretreatment [35]. NAC is able to inhibit the NF-κB translocation to the nucleus, as well as the p38 MAPK phosphorylation, and both of these signaling pathways can regulate the NRF2 expression [19,75]. In line with this, benzo[a]pyrene-induced NF-κB p65 and IκBα phosphorylation, and a nuclear translocation of NF-κB p65 and p50 subunits, were alleviated by a NAC pretreatment [49].

Both IL-8 and RANTES production are induced by diesel exhaust particles in HBECs, and p38 MAPK phosphorylation plays a critical role; NAC pretreatment not only decreased the number of phosphorylated tyrosine and threonine of p38 MAPK, but also consequently decreased the concentrations of the IL-8 and RANTES [28]. RANTES may also play an important role in AHR. 

The MAPK signaling pathway consists of three signaling pathways: ERK, JNK, and p38, which transduce a signal from the membrane to the nucleus [43,78]. A study by Wang et al., using both cellular (HBECs and animal (mice) models, showed that pretreatment with NAC prior to the exposure to particulate matter blocked the ERK, JNK, and p38 phosphorylation, suggesting the role of ROS in the MAPK activation [43]. As a consequence, the NF-κB activation was significantly inhibited, and the inflammatory response was attenuated, with a reduced COX-2 expression (both gene and protein level) and the expression of IL-1β, IL-6, IL-8, and MMP-9, as well as decreased concentrations of IL-1β, IL-6, IL-8, and MMP-9 in mouse bronchoalveolar lavage fluid (BALF) [43].

Oxidative stress

Oxidative stress appears to play a crucial role in the interaction between RSV and air pollution, as the generation of ROS has been suggested as a plausible explanation for various pathways, via a disrupted epithelial barrier, a decreased apoptosis, or an increased inflammatory response; the use of NAC is able to stop or reduce the detrimental effects of oxidative stress [37,45,72]. A model of RSV infection with HBECs (BEAS-2B) showed that RSV induced the production of ROS and MDA, and the SOD activity and the intracellular glutathione-to-glutathione disulfide (GSH/GSSG) ratio [25]. NAC treatment counteracted the production of ROS and MDA, and increased the superoxide dismutase (SOD) activity and the GSH/GSSG ratio, thus attenuating the pathological effects of the RSV infection [25]. Similarly, an in vitro model of diesel exhaust exposure showed that NAC has the ability to restore the intracellular GSH/GSSG ratio, counteract both lipid and protein oxidation, and suppress the expression ofHO-1, which is induced in response to oxidative stress [46]. In both human and animal models, particulate matter was shown to induce the ROS generation in a dose- and time-dependent manner, and NAC pretreatment was able to block the ROS production [43]. Also, PM2.5-induced ROS activation in a cellular model of HUVECs, and this effect was reduced by NAC [79]. Exposure to concentrated ambient particles induced oxidative stress, as measured by an increase in thiobarbituric reactive substances (TBARS, one of the typical indicators of oxidative stress) and oxidized proteins in rat lungs; NAC pretreatment reduced TBARS concentrations, but did not affect protein levels [37]. 

In addition to a decrease in oxygenation status, which can be seen both in RSV infection and under the influence of air pollution, hypoxia-reoxygenation stress causes damage to human epithelial cells via increased ROS production; this effect can be mitigated by NAC treatment [80]. One of the lesser-known but promising mechanisms of the cellular response to oxidative stress is a stress-induced production of the heat shock protein, HSPA6, which belongs to the Hsp70 family, and some viruses possess the ability to increase heat shock proteins at different stages; RSV was shown to increase both mRNA and protein expression of HSPA6, which were blocked by NAC administration [25].

PAMPs

Pathogen-associated molecular patterns (PAMPs) are recognized by pattern-recognition receptors (PRRs), which may be membrane-bound or located in the cytosol, and a pathogen-related immune response is then induced [81,82]. Toll-like receptors (TLRs) are the membrane-bound PRRs (together with C-type lectin receptors, CLRs); TLR3, TLR7, TLR8, and TLR9, recognize specific viral genomic components and play a crucial role in virus-induced signal transduction [81,82]. An investigation by Wang et al. proved a correlation between the TLR3 activation and oxidative stress in human lung epithelial A549 cells [44]. The RSV-infected cells showed increased concentrations of the hydroxyl free radical (·OH) and nitric oxide (NO), and a decreased total SOD activity; these effects were reduced by a pretreatment with NAC, and both hydroxyl free radical and NO levels exhibited a temporal dependence with the duration of the RSV infection [44]. In terms of the mRNA expression, an upregulation of TLR3 and NF-κB (with a corresponding increase in the protein expression of the TLR and phosphor-NF-κB-p65) and a downregulation of the interferon regulatory factor-3 (IRF3) and the superoxide dismutase 1 (SOD1) genes; a NAC pretreatment of RSV-infected cells reversed the mRNA and protein expression pattern, suggesting a possible regulatory role of oxidative stress in the TLR3 activation and an enhanced inflammatory response [44]. 

##### Inappropriate Mucus Secretion

MUC5AC

An inappropriate host response may also be mediated by an increased secretion and altered characteristics of mucus in the airways. The study by Mata et al. showed the ability of NAC to suppress the release of a mucin gene, MUC5AC. MUC5AC, a gel-forming mucin, is one of the major components of human mucus, and an enhanced effect of both RSV infection and air pollution (e.g., cigarette smoke) on mucus secretion may be expected [83]. Interestingly, overexpression of MUC5AC may be regulated by EGFR [84]. The aforementioned inhibited phosphorylation of EGFR exerted a positive effect in integrin β4 (ITGB4)-deficient mice, by reducing the MUC5AC overexpression and mucus secretion [84]. An integrin β4-mediated mechanism has been postulated to play a role in the pathogenesis of asthma. A decreased expression of ITGB4 is observed in asthmatic patients, and a cell line model of a ITGB4 deficiency (HBECs) revealed an increased phosphorylation of the EGFR after an RSV infection, while in humans, a lowered ITGB4 expression correlated inversely with the MUC5AC levels in throat swabs obtained from children infected with RSV [84]. In addition to its direct effects on mucus viscosity, MUC5AC also contributes to goblet cell metaplasia [83].

##### Structural Abnormalities

Goblet cell metaplasia and MUC5AC expression are increased under pathological conditions such as chronic rhinosinusitis or chronic obstructive pulmonary disease (COPD); in addition, cigarette smoke enhanced the pathological response, which was mainly mediated by the EGFR pathway [83,85]. A drop in the number of cilia-positive and beta-tubulin-positive cells was observed in the study by Mata et al. and was accompanied by ultrastructural abnormalities in the axonemal basal bodies in electron microscopy of NHBECs infected with RSV [35]. These alterations were mediated by a decrease in the expression of the *FOXJ1* and *DNAI2* genes, resulting in an overall decrease in the cilia activity [35]. Furthermore, metaplastic alterations were observed, probably mediated by an increased expression of MUC5AC and GOB5 (an ion channel that regulates the expression of *MUC5AC*), as well as an increase in the expression and release of IL-13 [35]. NAC was able to inhibit the upregulation in a dose-dependent manner and restore the epithelial functions [35]. In line with this, the study by Lei Chi et al. also showed that, in human HBECs (BEAS-2B cells), NAC treatment inhibited the phosphorylation of EGFR and the activation of MUC5AC [25].

Bronchoalveolar lavage (BAL) of rat lungs showed an influx of polymorphonuclear leukocytes evoked by concentrated ambient particles, accompanied by histopathological changes: thickening of bronchiole vessels, bronchiolar inflammation, and lung edema [37]. NAC pretreatment reversed the polymorphonuclear leukocyte influx and the animals showed no histological changes [37].

Pretreatment with NAC reversed the particulate-matter-induced inflammatory cell infiltration around the PM deposition in the lungs of mice, reduced the number of neutrophils and macrophages in BALF, and improved the lung injury scores [43].

Emphysema

Data on the attenuation of air pollution-induced emphysema by NAC are conflicting. A rat model of smoking-induced COPD studied by Cai et al. revealed advanced emphysema, which was attenuated by NAC; both the mean linear intercept (MLI) and the destructive index (DI) were lower in rats treated with NAC [23]. Importantly, NAC reduced the apoptotic index (AI) of alveolar septal cells (assessed by terminal deoxynucleotidyl transferase dUTP nick-end labelling (TUNEL) assay) and reversed a decrease in the expression of the vascular endothelial growth factor (VEGF) and its receptor protein (VEGFR2), with AI inversely correlating with the VEGF levels [23]. 

Histopathological changes in mice induced by benzo[a]pyrene include the thickening of the airway wall with a reduction in MLI, and an increase in DI; all were alleviated by NAC pretreatment, and overall lung pathological scores were improved [49]. 

On the other hand, a study by March used a murine model of cigarette smoke exposure, and although an increase in MMP-2 and MMP-9 activity was observed and emphysema was induced by exposure, NAC treatment failed to mitigate the severity of emphysema [32]. Another murine model investigated ozone-induced COPD and compared a preventive versus a therapeutic administration of NAC [30]. NAC delivered before exposure was able to reduce the number of macrophages in BAL and in the airway smooth muscle (ASM) mass, while the therapeutic administration was able to reduce the ASM mass and the number of apoptotic cells, but the emphysema was not reversible [30]. The mechanism of the emphysema is complex, and matrix metalloproteinases, particularly MMP-9 and MMP-12, are of interest; activation of the IL-17 → MCP-1(Ccl-2) → MMP-9 → MMP-12 axis appears to be critical for the formation of emphysema [86,87,88]. Another proposed mechanism of emphysema formation is apoptosis, although the possible protective role of NAC remains to be verified [30,88]. In a model of particulate matter exposure, a reduced expression and reduced levels of MMP-9 were observed in HBECs and BALF after NAC pretreatment [43].

DNA damage and senescence

Oxidative stress induces the damage of DNA and may trigger cellular senescence (with proliferative arrest), which has also been shown in the case of RSV infection in HEp-2 and A549 cells [33]. The observed increase in the expression of the markers of DNA damage and proliferation arrest were the result of the increased production of ROS, at least partially derived from mitochondria [33]. ROS cause (directly or as an effect of repair) DNA double-strand breaks (DSBs), which may be reversed by the use of antioxidants; interestingly, NAC substantially decreased the number of damaged nuclei, but its protective effect was only observed in the case of de novo DNA damage foci, with virtually no effect on the previously existing ones [33]. While a moderate reduction in the RSV titers was achieved with NAC treatment, a much stronger effect was observed when NAC was used during the RSV adsorption, suggesting a possible role for the antioxidants in RSV cell entry, surface attachment, or viral integrity [33]. The detrimental effects on DNA may extend beyond the period of infection; a murine model proposed by Martinez et al. not only demonstrated a DNA damage and senescence during infection (severe damage in 23% of cells and in 30% of columnar bronchiolar epithelia, as well as a moderate DNA damage in 30% of alveolar epithelia), but the alterations were present in 18% of bronchiolar epithelia when the RSV was cleared (at day 11 and 30 post-infection), which suggests a possible role for cell aging and tissue remodeling in the long-term sequelae of an RSV infection [33]. 

A similar effect on ageing and cellular senescence has been suggested for various air pollutants, including particulate matter (PM2.5, PM10), NO, or cigarette smoke, and oxidative stress seems to be the most plausible mechanism of interaction [64,89,90,91]. The study by Kang et al. verified the influence of TiO_2_-NP, revealing an increase in ROS production, activation of the p53-mediated response to ROS, and DNA damage in peripheral blood lymphocytes (obtained from a healthy donor) [29]. The effects on the ROS production and the DNA breakage were reversed by a pretreatment with NAC [29]. On the other hand, the study by Wan et al. revealed that TIO_2_ nanoparticles, which are widely used in studies modeling lung effects of environmental nanoparticles, did not cause an increase in the ROS generation in the exposed A549 cells, and the consequences of the ROS generation, i.e., DNA damage, the phosphorylation of histone H2AX (γ-H2AX), Rad51, and p53, or the phosphorylation of ataxia telangiectasia mutant were seen after the exposure to cobalt nanoparticles and reversed by NAC. However, we did not include cobalt nanoparticles as a model of air pollution [42]. Nonetheless, in the case of TiO_2_-NP, these incompatibilities may be at least partially explained by the aforementioned study by Wen et al., where no increased ROS generation was found after the TiO_2_-NP exposure but the catalase activity was significantly increased [45]. Thus, a cumulative or additive effect of air pollution and RSV might be expected, and antioxidants could play a protective role against both DNA damage and cellular ageing.

#### 3.3.4. Airway Hyperresponsiveness (AHR)

Data on the mechanisms of NAC in RSV- or air-pollution-induced AHR are rather scarce, although the increase in inflammatory mediators shown in other studies seems to justify an expectation of AHR following both RSV and air pollution exposure, with a likely interference and amplification of the pathways. A mixed exposure to RSV and air pollution may result in increased levels of RANTES, MCP-1, and NGF, and decreased levels of the leukemia inhibitory factor (LIF), leading to AHR [11]. The mechanisms consist of a few interfering pathways; in brief, the MCP-1 (and the MIP-1a, e.g., induced by carbon black and RSV) are chemoattractants for eosinophils, which release RANTES, leading to an increased responsiveness [11]. Of note, the studies cited above demonstrated the efficacy of NAC in reducing levels of both RANTES and MCP-1 in the models of RSV exposure and air pollution exposure (diesel emissions and benzopyrene, respectively), which may further result in a reduced or absent hyperresponsiveness. 

A murine model of O3 exposure to AHR assessed by acetylcholine challenge was reversed by a therapeutic administration of NAC, but not by a preventive administration of NAC [32]. The authors postulate that while NAC does not effectively prevent from oxidative stress, the therapeutic administration of NAC may be more effective at later stages of oxidative stress, thereby reducing the AHR [32].

## 4. Discussion

The use of a systematic review approach revealed only a small number (28) of papers that have addressed the topic of molecular mechanisms of NAC action against RSV infection and/or air pollution. NAC is a well-known drug, with a long history of use; its mechanisms and safety profile are well established, and the interest in the clinical use of NAC has recently increased with a large number of clinical trials focusing on viral infections. Similarly, the issue of air pollution and its influence on both morbidity and disease severity is gaining more attention, with studies suggesting that certain viruses (especially RSV and influenza) benefit from air pollution, but there is a limited number of studies that re-evaluate the exact mechanisms behind the NAC effects in patients exposed to a viral infection and/or air pollution. A better understanding of the underlying and, in particular, coincidental pathways may focus the attention on the use of specific substances. In this respect, NAC appears to be one of the most promising drugs. 

We identified the following major gaps in our knowledge in the analyzed areas:-RSV: There is limited interest in the use of NAC for RSV infections, which could be an important direction for future studies, given the lack of registered, effective, widely used, and available drugs for RSV-caused diseases, and the lack of targeted anti-RSV treatment;-Air pollution: There is a lack of a detailed analysis of interventions for specific air pollutants; while the majority of studies focused on TiO_2_-NP or cigarette smoke, very little attention is paid to the six air pollutants acknowledged by the WHO as the most important ones (PM_2.5_, PM_10_, O₃, NO₂, SO₂, and CO);-Mechanisms:oThe choice of the best route of administration needs to be verified;oA key role for the timing, duration, and dose of NAC might be expected. While some studies have tested the effects of different NAC doses, there seems to be another important and underestimated problem—the effects of timing (NAC pretreatment versus treatment)—where head-to-head comparisons would give the most accurate answers. Little is known about the optimal duration of NAC administration; oThe lack of focus on long-term sequelae of RSV and/or air pollution; the studies tend to focus on the immediate effects, although from a clinical perspective, long-term complications are more important.

All three main routes of administration are available for NAC treatment in humans; NAC may be administered orally, intravenously, or by inhalation [2]. The series of experiments in human type II pulmonary epithelial cell line derived from human lung adenocarcinoma (A549) showed a dose-dependent antiviral activity and cytotoxicity of NAC [42]. While a three-fold decrease in the number of RSV-infected cells was observed as an effect of pretreatment with 1 mM NAC, no significant changes were observed after 0.1 mM NAC, and no infected cells were observed after 10 mM NAC [47]. However, while the lower and medium concentrations of NAC did not induce any morphological changes in the epithelium, the highest dose resulted in severe cell injury, observed as an increase in the cytoplasmic/nuclear ratio, wider interstitial and intercellular spaces, and worsened adhesion [42]. 

Dose-dependency was also verified in another study with adenocarcinomic human bronchoalveolar basal epithelial cells (A549); previously, it had been suggested that lower doses (600 mg orally) would elicit antioxidant effects, while higher doses would result in anti-inflammatory effects [92,93,94]. The study tested four different doses: 16 μM (equivalent to 600 mg orally), 35 μM (equivalent to 1200 mg), 1.6 mM, and 5 mM (most commonly used in in vitro concentrations); prolonged NAC treatment exerts both antioxidant and anti-inflammatory effects, while acute high-dose treatment has strong antioxidant and anti-inflammatory effects [92].

Recently, much attention has been paid to NAC in COVID-19 due to the growing evidence of the role of the “cytokine storm” in its pathophysiology. A number of publications have suggested a role for NAC in the treatment of COVID-19, leading to an increasing number of clinical trials on the use of NAC in SARS-Co-V-2 infected patients [22,95]. Different strategies of NAC administration were proposed at the beginning of the pandemic, some of which suggested adapting the route of administration to the stage of the disease; while oral NAC was advised in the early stages of COVID-19 or even as an immunomodulatory prophylactic measure, the inhaled route was advised for more severe cases (without airway obstruction), and IV administration was advised as an adjunctive therapy in those who developed pneumonia and/or dyspnea [22]. A potential therapeutic role of NAC in COVID-19 needs to be thoroughly verified by a number of ongoing trials, and the example of COVID-19 shows a problem related to the choice of the most appropriate route of administration, timing, and dosage of the NAC. 

Clinical trials of NAC in humans with an RSV infection have shown some promising results. The aforementioned study by Zhou et al. included 126 COPD patients, and showed that both the use of NAC or terbutaline sulfate, and especially the combined treatment with NAC and terbutaline sulfate, raised the clinical scores of the patients [72]. An improved blood gas profile was observed (higher oxygenation index (OI) and blood oxygen saturation (SaO_2_), together with a lower partial pressure of carbon dioxide- PaCO_2_), alongside better results in functional studies, i.e., a forced vital capacity, maximum mid-expiratory flow rate, and peak expiratory flow [72]. There was also a progress in the COPD assessment test and the 6-min walk distance, with no differences in the incidence of adverse effects [72].

A study of 26 healthy volunteers confirmed that individuals with airway hyperresponsiveness benefited from oral supplementation with NAC prior to exposure to diesel exhaust containing 300 ug/m^3^ of PM_2.5_; whereas diesel exhaust provoked a marked airway hyperresponsiveness in individuals prone to AHR; this effect was abolished when they were pretreated with NAC [96]. 

A prospective randomized controlled trial of infants and children aged 2–24 months was performed in Lahore, Pakistan, and published in 2014; the patients hospitalized due to viral bronchiolitis were randomly assigned to one of two study arms and received 20 mg nebulized N-acetylcysteine or 2.5 mg nebulized salbutamol [97]. A total of 100 children were enrolled, and although no data were presented on the etiological factors of bronchiolitis, it is reasonable to assume that a significant number of patients were infected with RSV, as RSV is the most common etiological agent of acute bronchiolitis in children, and the vast majority of recommendations do not advise etiological studies, even in the case of hospitalization [98]. Patients who received NAC had better clinical severity scores on the subsequent assessment days (better on day 5 than on day 3) and a trend (without statistical significance) towards shorter length of stay, leading the authors to conclude that such therapy may be effective in bronchiolitis [97]. 

Currently, the number of registered NAC clinical trials is high, reaching 419 on clinicaltrials.gov and 75 on EudraCT, with some trials focusing on viral respiratory diseases, mostly on the SARS-CoV-2 infection, but also on acute bronchiolitis or influenza [99,100].

The present scoping review has certain limitations. Firstly, the small number of studies included, which is due to the small number of studies conducted rather than a shortcoming of the review process itself (we used a systematic approach to reduce the risk of any search bias). We have presented (and treated equally) the results from different models, although the reader needs to be aware of the advantages and disadvantages of each model, and the possible differences in molecular responses; however, the differences are likely to be related to the magnitude of the response and its timing, rather than to the mechanism itself (although this cannot be excluded). Another limitation of the study is the decision to present mechanisms for different air pollutants in a single review, as the pathways induced may differ significantly between the air pollutants. There are other important differences in the host response to RSV, air pollution and NAC—namely, patient-dependent differences—especially in terms of the age groups (e.g., infants versus elderly patients), or the question of dosage, timing of the NAC administration, or duration of the treatment, and clinical studies should provide the answers to the practical use of NAC; nevertheless, molecular studies are generally indispensable as a first step before the clinical application of any substance.

## 5. Conclusions

This review has potentially identified the common molecular mechanisms of interest that are used by NAC in the mitigation of both RSV infection and the effects of air pollution. Data are scarce, although beneficial multidirectional effects may be seen at different stages of the RSV infection, from blocking the viral entry and reducing the viral load to averting the prolonged disease course, or long-term sequelae related to an inappropriate or exaggerated host response (enhanced inflammation, increased mucus viscosity, AHR, DNA damage, and/or cellular senescence). 

Further research should be undertaken to investigate the effects of NAC during an RSV infection, with particular focus on the most important air pollutants and the molecular mechanisms they use. With this knowledge, targeted prophylactic or treatment options may become available. 

## Figures and Tables

**Figure 1 ijms-25-06051-f001:**
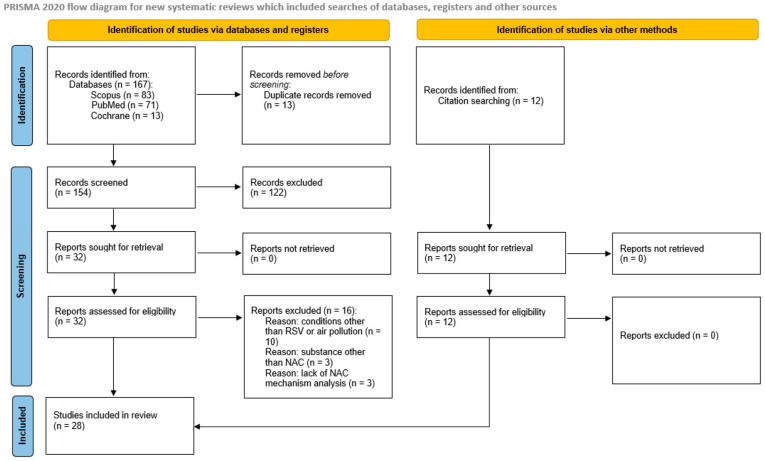
PRISMA flow diagram.

**Figure 2 ijms-25-06051-f002:**
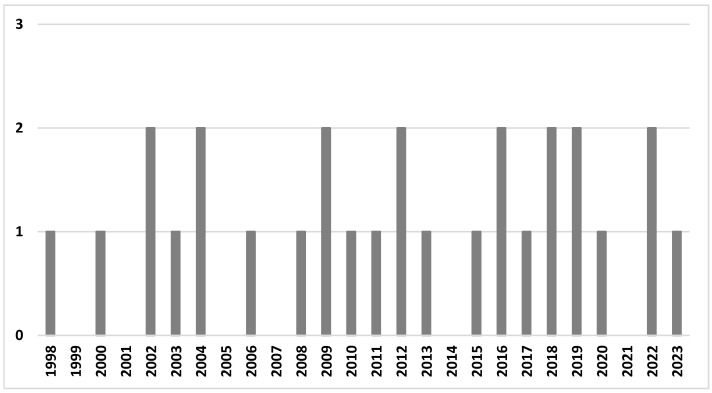
The number of studies in the review by year of publication.

**Figure 3 ijms-25-06051-f003:**
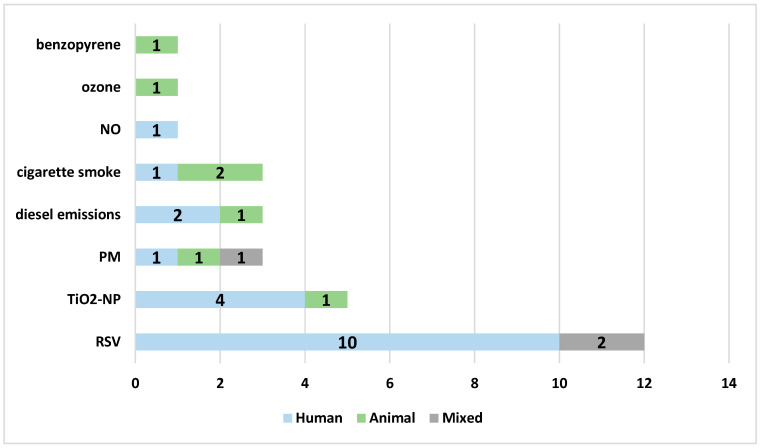
The models of the studies according to the specific variables analyzed in the review: blue—human cellular model, green—animal model, gray—mixed model. Abbreviations: RSV—respiratory syncytial virus, TiO_2_NP—titanium dioxide nanoparticles, PM—particulate matter, NO—nitric oxide.

**Figure 4 ijms-25-06051-f004:**
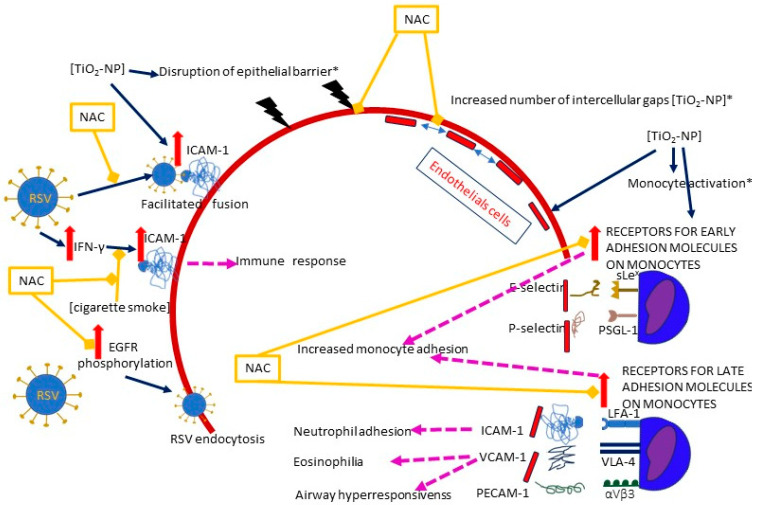
A diagram of the effects of NAC on the plausible mechanisms of RSV entry facilitated by air pollution and/or RSV itself. Solid dark blue arrows indicate the directions of the activity of RSV and/or air pollutants (the particular air pollutants are indicated in square brackets), with the directions of the effects marked by red or blue arrows (for increase and decrease, respectively); dashed pink lines show the effects of the activation of specific pathways; blind yellow arrows indicate the sites where NAC inhibits the mechanisms induced by RSV and/or air pollution; the solid red line represents the epithelium, and the dashed red line represents the endothelium. The pathways in which oxidative stress plays a significant role are marked with an asterisk (*). The figure is based on a literature search and is simplified for illustrative purposes only. Abbreviations: TiO_2_-NP—titanium dioxide nanoparticle, ICAM-1—Intercellular adhesion molecule-1, IFN-γ—interferon gamma, VCAM-1—vascular cell adhesion molecule 1, PECAM-1—platelet endothelial cell adhesion molecule-1, EGFR—epidermal growth factor receptor, sLex—E-selectin ligand, PSGL-1—P-selectin ligand, LFA-1—Lymphocyte function-associated antigen 1 (ICAM-1 ligand), VLA-4—very late antigen-4 (integrin α4β1, VCAM-1 ligand), αVβ3—PECAM-1 ligand.

**Figure 5 ijms-25-06051-f005:**
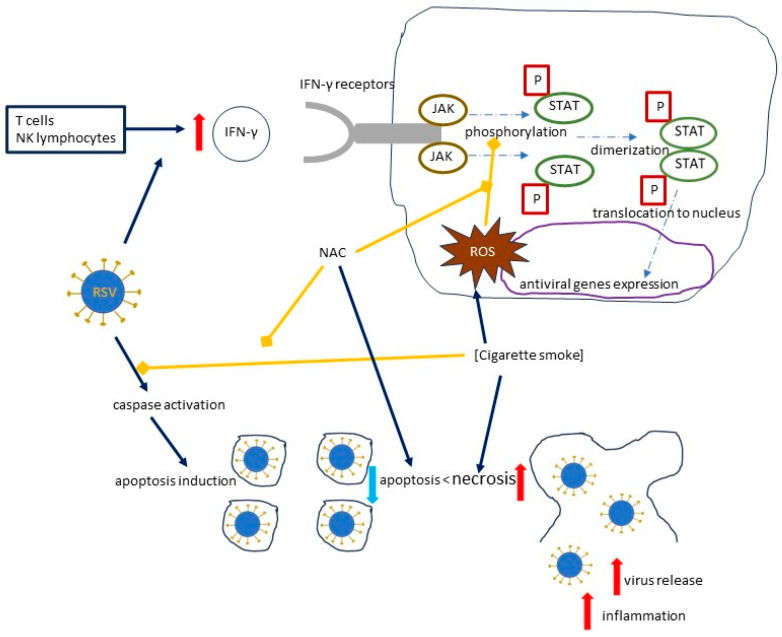
The effects of cigarette smoke on the RSV load, focusing on the mechanisms induced by cigarette smoke and inhibited by NAC. Solid dark blue arrows indicate the directions of the activity of human immune cells, RSV, and cigarette smoke, with the directions of the effects marked by red or blue arrows (for increase and decrease, respectively); dotted and dashed blue arrows indicate the directions of signal transduction and its effects, while blind yellow arrows indicate the inhibitory effect of NAC, which inhibits the mechanisms induced by cigarette smoke, and by cigarette smoke, which inhibits caspase activation; curved solid black line represents a cell. Abbreviations: IFN-γ—Interferon gamma; P—phosphoryl group; JAK—Janus kinase; STAT—signal transducer and activator of transcription; ROS—reactive oxygen species.

**Figure 6 ijms-25-06051-f006:**
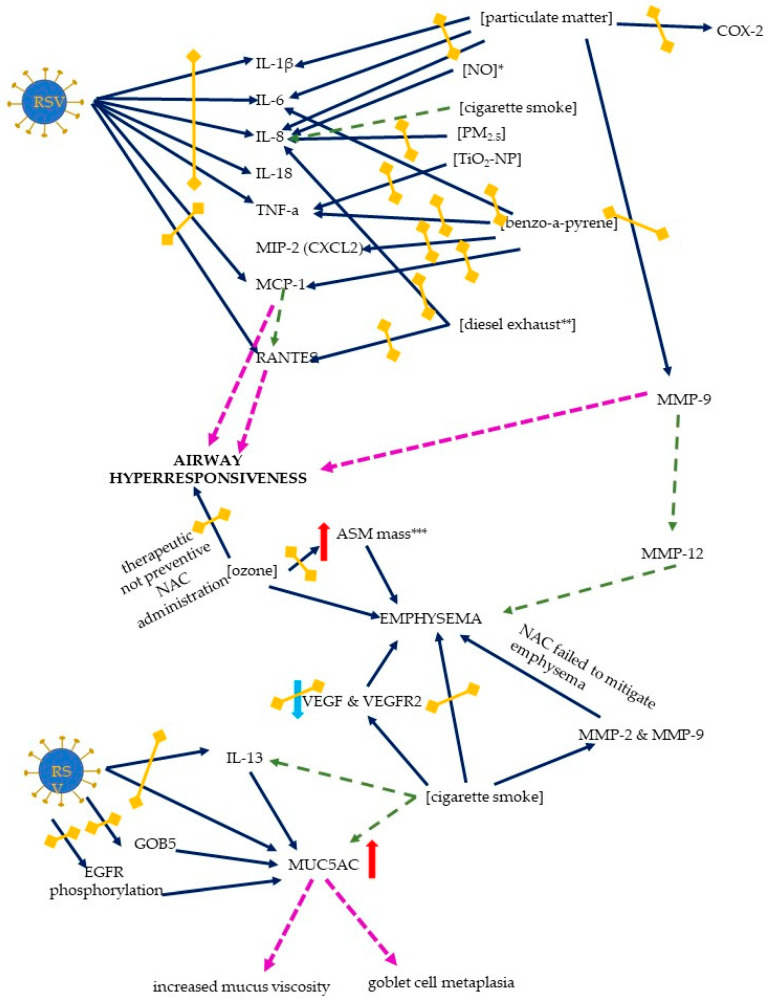
The mechanisms of host response to an RSV infection exerted by the RSV and air pollutants. The exaggerated response includes an inflammatory response, airway hyperresponsiveness, emphysema formation, and mucus changes. Solid dark blue arrows indicate the directions of action of the RSV and/or air pollutants (the particular air pollutants are indicated in square brackets), with the directions of the effects indicated by red or blue arrows (for increase and decrease, respectively); in the case of proinflammatory chemokines and cytokines, the red arrows (only increase was seen) are not shown for better readability. The effects of the activation of the specific pathways are indicated by pink or green dashed lines (the latter are used to signal the results that are not directly derived from this scoping review); double-headed blind yellow arrows indicate the sites where NAC inhibits the mechanisms induced by RSV and/or air pollution. Abbreviations: IL-1 β, -6, -8, -13, and -18—interleukin-1 β, -6, -8, -13, and -18 (respectively), TNF-α—tumor necrosis factor alpha, MIP-2—macrophage inflammatory protein-2, MCP-1—monocyte chemoattractant protein-1, RANTES—regulated upon activation, normal T cell expressed and secreted, ASM—airway smooth muscle, VEGF—vascular endothelial growth factor, VEGFR—receptor for vascular endothelial growth factor, EGFR—epidermal growth factor receptor, MUC5AC—mucin-5AC, MMP-2, -9, and -12—matrix metalloproteinase 2, 9, and 12 (respectively), TiO_2_-NP—titanium dioxide nanoparticle, PM2.5—particulate matter 2.5, NO—nitric oxide, COX-2—cyclooxygenase 2. * NAC does not influence IL-8 levels, ** one of the studies showed only a trend towards reducing IL-8 levels by NAC, *** NAC reduces ASM mass, but not emphysema.

**Table 1 ijms-25-06051-t001:** Eligibility criteria.

Inclusion Criteria	Exclusion Criteria
Original research	Review study
Publication date: 1998–2023	Publication before 1998
Accepted after a peer-review process	Lack of peer-review (e.g., gray literature, preprints, etc.)
Animal, human cellular, or mixed model	Lack of a model (e.g., epidemiological studies, clinical studies)
Analysis of NAC molecular mechanism(s) of action	Lack of analysis of NAC molecular mechanisms
RSV infection or air pollution model	Lack of analysis of RSV or the influence of air pollution
	Only maternal/in utero exposure analysis
	NAC mechanism in condition other than RSV or air pollution
	Study on combined effects only (e.g., various etiological agents, or allergy models)
	Analysis of effects other than in the respiratory tract

**Table 2 ijms-25-06051-t002:** Studies finally included in the scoping review.

	First Author, Publication Year	Model	Air Pollutant	RSV	Material/Subjects	Country
1	Cai, 2009 [23]	A	cigarette smoke		male Sprague–Dawley rats	China
2	Carpenter, 2002 [24]	H		+	A549 human type II lung carcinoma cell line	USA
3	Chi, 2022 [25]	H		+	human bronchial epithelial cells BEAS-2B	China
4	Dick, 2003 [26]	A	four different ultrafine particles (carbon black, cobalt, nickel, and titanium dioxide)		male Wistar rats	UK
5	Groskreutz, 2009 [27]	H	cigarette smoke extract	+	primary human tracheobronchial epithelial	USA
6	Hashimoto, 2000 [28]	H	diesel exhaust particles		transformed human bronchial epithelial cell line BET-1A	Japan
7	Kang, 2008 [29]	H	TiO_2_-NP		peripheral blood lymphocytes from a healthy female adult donor	South Korea
8	Li, 2013 [30]	A	ozone		C57/BL6 mice	UK, China, USA
9	Li, 2018 [31]	H/A		+	human laryngeal epithelial cell line HEp-2 and HEK 293T cells and BALB/c female mice	China/USA
10	March, 2006 [32]	A	cigarette smoke		male and female A/J mice	USA
11	Martinez, 2016 [33]	H		+	HEp-2 and A549 cells	Spain
12	Mastronarde, 1998 [34]	H		+	alveolar epithelial cells A549	USA/Japan
13	Mata, 2011 [19]	H		+	human pulmonary epithelial cell line A549	Spain
14	Mata, 2012 [35]	H		+	primary normal human bronchial epithelial cell (NHBEC)	Spain
15	Modestou, 2010 [36]	H	cigarette smoke extract	+	human trachea and bronchial samples, primary human tracheobronchial epithelial cells	USA
16	Rhoden, 2004 [37]	A	concentrated ambient particles (CAPs)		adult male Sprague-Dawley rats	USA
17	Rueda-Romero, 2016 [38]	H	TiO_2_-NP		human cell line U937 as a monocyte cell model and HUVECs as a model for endothelial cells	Mexico, Poland, Sweden
18	Smallcombe, 2020 [39]	H/A	TiO_2_-NP	+	immortalized human bronchial epithelial cells; C57BL/6 mice	USA
19	Sparkman, 2004 [40]	H	nitric oxide		NCI-H441 cells, a human lung adenocarcinoma cell line of bronchiolar (Clara) cell lineage, and BEAS-2B cells, an SV40 transformed human bronchial epithelial cell line	USA
20	Vaughan, 2019 [41]	H	diesel emission		primary human bronchial epithelial cells (pHBEC) from patients with/without COPD	Australia
21	Wan, 2012 [42]	H	nano-sized cobalt (nano-Co) and titanium dioxide (nano-TiO_2_)		human lung epithelial cell lines A549	USA
22	Wang, 2017 [43]	H/A	urban particulate matter 1649b		human bronchial epithelial cells (HBECs) Male C57 mice mice	China
23	Wang, 2018 [44]	H		+	human lung adenocarcinoma alveolar basal epithelial cell line A549 and laryngeal epithelial carcinoma HEp-2 cell line	China/USA
24	Wen, 2019 [45]	H	nanoparticles including gold (Au), platinum (Pt), silica (SiO_2_), titanium dioxide (TiO_2_), ferric oxide (Fe_2_O_3_), oxidized multi-walled carbon nanotubes (MWCNTs)		primary human umbilical vein endothelial cells (HUVECs)	China
25	Whitekus, 2002 [46]	A	diesel exhaust particles		murine macrophage cell line, RAW 264.7 cells	USA
26	Wong, 2023 [47]	H		+	human type II pulmonary epithelial cell line A549	Malaysia
27	Yan, 2015 [48]	H	PM_2.5_		human bronchial epithelial cell line (BEAS-2B cells) and human macrophage-like cell line (THP-1 cells)	China
28	Zhao, 2022 [49]	A	benzo[a]pyrene		C57BL/6J male mice	China

## Data Availability

Data supporting the reported results are available on request from the authors.

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
