# Peer review of "Molecular Mechanisms of N-Acetylcysteine in RSV Infections and Air Pollution-Induced Alterations: A Scoping Review"

_ijms, 2024, doi:10.3390/ijms25116051_

Round 1

Reviewer 1 Report (Previous Reviewer 3)

Comments and Suggestions for Authors

This manuscript is a comprehensive review of the published data on the molecular mechanisms of the N-acetylcysteine in RSV infections and air pollution-induced alterations. The authors have done a great job of reviewing the current literature and summarizing the ideas about the effects of the drug on these two processes. The original submission was significantly revised and re-organized, and current version can be accepted for publication in IJMS

Author Response

Dear Reviewer,

Thank You very much for Your courteous opinion, we are glad that the changes we made meet Your vision of this article. In fact, thanks to the suggestions and corrections made during the review process, the article has been changed and is now, in our opinion, much more readable and understandable.

Thank You for all Your help!

Reviewer 2 Report (New Reviewer)

Comments and Suggestions for Authors

In their manuscript entitled “Molecular mechanisms of N-acetylcysteine in RSV infections and air pollution-induced alterations: a scoping review", Wrotek et al., discuss the impact of N-acetylcysteine (NAC) on respiratory syncytial virus (RSV) infections and air pollution-induced pulmonary alterations. The study aims to review literature identifying molecular mechanisms through which NAC treatment may affect pulmonary alterations due to RSV infections and air pollution.

The authors provide a decent background in the introduction, where N-acetylcysteine is explored for its mucolytic, antioxidant, and anti-inflammatory properties and RSV is highlighted as a major cause of lower respiratory tract infections, with air pollution exacerbating RSV incidence and severity.

The authors then performed a systematic search to identify 28 relevant studies focusing on RSV infection, air pollutants and NAC treatment in human cellular, animal, and mixed models. The authors have done a splendid job at reviewing the literature, and have reported the relevant findings in a highly organize manner, enriched with detailed confirmed or suspected mechanisms by which NAC treatment may be acting. The authors highlight and discuss the following mechanisms by which NAC treatment can be beneficial and serve protective functions.

1  NAC improving epithelial barrier function and reducing viral entry.

2 NAC blocking RSV-induced phosphorylation of EGFR, inhibiting cell entry.

3  NAC preventing virus release from infected cells.

4  NAC reversing shift from necrosis to apoptosis induced by cigarette smoke.

5 NAC restoring IFN-γ-induced antiviral gene expression inhibited by RSV.

6  NAC attenuating oxidative stress-induced DNA damage and cellular senescence.

  Overall, the article is well written, easy to follow, and contains a detailed review of relevant studies and potential molecular mechanisms. This review emphasizes the potential of NAC in mitigating the effects of RSV infection or air pollution induced pulmonary alterations, and summarizes important information for public health, basic and translational research, and will likely be of interest to the scientific community and public alike.

I only have a few minor comments:

11. The authors exclude studies published prior to 1998 but do not provide a good reason for doing so. While older literature may not always hold to the same molecular rigor as more recent studies, they may provide important data and relevant historical context that can enrich the manuscript. Sometimes, important data can be found on older literature that can make a big difference when considered along more recent information

22. In Figure 1, the PRISMA flow diagram, it would be helpful to break down the initial Records excluded from screening (n=122) by category of exclusion criterion. It appears that the authors originally meant to add this information to the figure legend, but I was not able to find this information in the manuscript.

33. There are many areas where the authors refer to particular studies simply by the first author’s name even though many of them are from multiple author studies. It is best practice to use (First author, et al.,) for these, to acknowledge the contributions the other authors made as well.

44. The number of studies reviewed is relatively small, but the authors acknowledge that as a limitation, and this is simply a matter of there simply not being many studies that fit the inclusion criteria and should not stand in the way of publication

Author Response

Dear Reviewer!

We would like to express our gratitude for your kind review and constructive comments that will improve this manuscript! We would like to respond directly to all the points you raised (we italicised your comments to make them easier to read):

  1. The authors exclude studies published prior to 1998 but do not provide a good reason for doing so. While older literature may not always hold to the same molecular rigor as more recent studies, they may provide important data and relevant historical context that can enrich the manuscript. Sometimes, important data can be found on older literature that can make a big difference when considered along more recent information

I could not agree more! However, the reason for limiting the search period was based on the huge changes that have taken place in molecular biology in recent years, not on molecular rigor, which I have the impression was of higher quality 30 years ago than it is now. Nevertheless, the capabilities have changed significantly, and since a relatively small number of included papers was expected, the risk of bias due to conflicting results had to be excluded if, for example, a newer, more sensitive method could detect more detailed differences (e.g., in chemokine concentrations) that would be statistically significant.

We would like to emphasize that the search period is quite long (in fact, it is the last quarter century), but I fully agree that some important data might be found in "older" papers. That is why we also cited references from the early nineties.

An appropriate comment has been added to the Materials and Methods section.

  1. In Figure 1, the PRISMA flow diagram, it would be helpful to break down the initial Records excluded from screening (n=122) by category of exclusion criterion. It appears that the authors originally meant to add this information to the figure legend, but I was not able to find this information in the manuscript.

Yes, indeed, although it is not typical to include this information in the PRISMA flow diagram (at this stage), but we do include the information in the text.

  1. 33. There are many areas where the authors refer to particular studies simply by the first author’s name even though many of them are from multiple author studies. It is best practice to use (First author, et al.,) for these, to acknowledge the contributions the other authors made as well.

Sorry for that, it could be perceived as an underestimation, which is not our goal at all. We have changed all citations with special attention to the number of authors!

  1. The number of studies reviewed is relatively small, but the authors acknowledge that as a limitation, and this is simply a matter of there simply not being many studies that fit the inclusion criteria and should not stand in the way of publication

This is what hurts the most! The lack of abundant data on the topic! We believe that the scientific community needs and deserves more studies on this topic and hope that the future will definitely bring more data.

Best regards,

The authors

This manuscript is a resubmission of an earlier submission. The following is a list of the peer review reports and author responses from that submission.

Round 1

Reviewer 1 Report

Comments and Suggestions for Authors

The article is generally devoted to the mechanism of the antiviral action of NAC, which is active against RSV. Firstly, I don't see anything new in this manuscript. I don't understand the scientific intent. NAC have been studied from 1950 years. The authors simply presented the results of 28 publications. The connection between RSV infection and air pollution is not obvious to me. Is there a connection between flu infection and air pollution?

The conclusion is unclear.

The quality of the pictures leaves much to be desire

Reviewer 2 Report

Comments and Suggestions for Authors

I enjoyed reading this manuscript that reviews extremely well a variety of written review of N-acetylcysteine in RSV infections focuses on reviewing studies that involve air pollutants, such as ozone, and tobacco smoke.  The emphasis on the molecular mechanisms involved in the interaction of N-acetylcysteine with RSV is appropriate for this journal.

There are several papers I am aware of that use murine and/or primate models to evaluate the effects of side-stream smoke, ozone, and diesel exhaust on the lung, particularly with emphasis on effects of these pollutants on induction of allergic/asthmatic sensitization. While RSV is not a part of these studies, it is true that RSV infection is also associated with enhanced allergy/asthma. Incorporation of these studies might yield some interesting links. 

I commend the authors for putting this data together; it should be quite useful for future project and ultimately enhancing RSV treatment options.

Reviewer 3 Report

Comments and Suggestions for Authors

This manuscript is a comprehensive review of the published data on the molecular mechanisms of the N-acetylcysteine in RSV infections and air pollution-induced alterations. The authors have done a great job of reviewing the current literature and summarizing the ideas about the effects of the drug on these two processes, but as presented, it is very difficult to absorb the information presented and major revision of the manuscript is required before it can be published.

1.       The authors previously published a similar review focusing on the molecular mechanisms underlying the interaction between RSV infection and air pollutants (DOI: 10.3390/ijms232012704). Such a review was logical and understandable. Here, however, the authors combined the effect of the NAC on the virus and on air pollutants in one review, but it is not clear why. It is possible to understand the desire to find the molecular mechanisms underlying the drug's effect on viral infection, enhanced by air pollutants. But it is not clear why the review should include separately papers indicating an effect on pollutants per se. This makes it very difficult to understand the material, since the main idea anyway revolves around RSV infection and the different stages of virus life cycle.

2.       The abstract is too long and it is difficult to grasp the main ideas of the review, which is buried under a huge number of terms and listing of many molecular targets for the NAC's action. The abstract should concisely present the main idea of the review and the authors' comments on the main patterns found.

3.       The review includes a lot of redundant information unrelated to the purpose of the study, e.g. in the lanes 89-94. Please make sure that all the introductory information refers to the scope of the review. Similarly, the concluding remarks should refer to the main paper topic only.

4.       The authors define models used in the studies as human, animal or cellular; but the models shown in the Table 2 as H (human) in reality refer to studies dealing with human cell lines. What is the difference between human and cellular models here?  It seems incorrect to say “human model” when studies are performed using human cell lines. “Human cellular model” should be used instead.

5.       The authors mention multiple times that the RSV entry may be facilitated by RSV itself. The RSV entry is a virus-defined process that involves particular stages of virus-host interactions, and this process can be facilitated by some external conditions, such as air pollutants. But how the virus can facilitate its entry by itself?

6.       Lane 779: respiratory alveolar epithelial cells (A549); Lanes 787-788: adenocarcinoma human bronchoalveolar basal epithelial cells (A549). Please explain these inconsistencies.

7.       The authors repeatedly provide definitions of some of the abbreviations in the text.

Comments on the Quality of English Language

   Serious English language correction is required, as many sentences are incoherent and in some places the meaning is simply not clear.